# Study on Runoff and Infiltration for Expansive Soil Slopes in Simulated Rainfall

**Wenkai Lei [1], Hongyuan Dong [1], Pan Chen [1], Haibo Lv [1,2], Liyun Fan [2] and Guoxiong Mei [1,\*]**

[1] College of Civil Engineering and Architecture, Guangxi University, Nanning 530000, China
[2] College of Architecture Engineering, Hezhou University, Hezhou 542600, China
[\*] Correspondence: 1710402003@st.gxu.edu.cn; Tel.: +86-771-3232-242

**Abstract:** In order to understand the hydrological process of expansive soil slopes, simulated rainfall experiments were conducted to study the effects of slope gradient and initial soil moisture content on runoff and infiltration for expansive soil slopes located in south China. The field program consisted of four neighboring slopes (70%, 47%, 32%, and 21%) instrumented by a runoff collection system and moisture content sensors (EC-5). Results from the monitored tests indicate that there was delay in the response of surface runoff. The runoff initiation time decreased with initial soil water content and increasing slope gradient. After the generation of runoff, the cumulative runoff per unit area and the runoff rate increased linearly and logarithmically with time, respectively. The greater the initial soil moisture content was, the smaller the influence of slope gradient on runoff. A rainfall may contribute from 39% to about 100% of its total rainfall as infiltration, indicating that infiltration remained an important component of the rainwater falling on the slope, despite the high initial soil water content. The larger the initial sealing degree of slope surface was the smaller the cumulative infiltration per unit area of the slope. However, the soil moisture reaction was more obvious. The influence of inclination is no longer discernible at high initial moisture levels. The greater the initial soil moisture content and the smaller the slope gradient, the weaker was the change of soil water content caused by simulated rainfall. The influence of initial soil moisture content and slope gradient on the processes of flow and changes of soil water content identified in this study may be helpful in the surface water control for expansive soil slopes.

**Keywords:** run off; infiltration; expansive soil slopes; slope gradient; initial soil moisture content

## 1. Introduction

Expansive clay is a special soil formed in the natural geological process with obvious swelling and shrinkage, distributed in more than 40 countries and regions [1]. The annual loss caused by the problems related to expansive soil in the world was as high as $15 billion [2]. The failure of expansive soil slopes was a common problem in the rainy season. Rainwater infiltrates into the unsaturated soil slope, leading to changes in suction and gradually evolving to the state that is harmful to the stability of the slope [3]. Chen et al. [4] analyzed the influence of water infiltration and fracture on the deformation of expansive soil slopes with different densities using centrifugal model tests and found that rainwater was responsible for the slope failure. Ng et al. and Zhan carried out artificial rainfall simulation tests on an 11 m high cut slope in a typical medium-plastic expansive clay [5] and studied the interaction between expansive soil and water [6]. The results demonstrated that rainwater infiltrates into the expansive soil slope and the soil expands by absorbing water, leading to changes in stress, deformation, permeability, and shear strength of the slope. Qi and Vanapalli [7] evaluated the coupling effect of swelling and hydraulic response on the stability of

surficial layer in a typical expansive soil slope and highlighted that the coupled hydro-mechanical behavior of expansive soil has an adverse effect on slope stability. Khan et al. [8] investigated the failure of a highway slope and concluded that both a fully softened condition and rainwater are the important factors affecting the deformation of the expansive soil slope.

Model tests and field investigation indicated that rainfall infiltration is an important inducing factor for the failure of an expansive soil slope. The expansive soil slope is extremely vulnerable to damage during the rainy days without proper protection and treatment [9]. Therefore, the key measure to prevent the failure of expansive soil slope is to cut off the change of water as much as possible. Appropriate drainage measures should also be taken to prevent the slope from fully humidifying [10]. Infiltration of a soil slope is controlled by several bio-physical factors including ground cover, soil hydraulic properties, rainfall intensity, soil surface features, and slope gradient [11]. Slope gradient is an important factor in the stability of expansive soil slopes, affecting the hydrological process of slope surface, the development of surface cracks, and landslide [12].

After rainwater falling on the slope, some of it flows along the slope and some of it infiltrates into the slope, corresponding to the runoff and infiltration of the slope, respectively. The geometric and physical characteristics can be taken into account to reduce the influence of rainfall on the slope. Robert E. Horton, an American expert in environmental soil science, founded the slope hydrology, and pointed out that the slope with poor permeability and steep slope was conducive to the runoff of rainwater that could not infiltrate into soil in time [13]. Expansive soil is a kind of overconsolidated clay with low permeability [14] and high shear strength [8] when the cracks are not obvious. If the appropriate measures can be taken to reduce the infiltration of rainwater and avoid the excessive change of soil moisture content on the slope, the influence of rainfall on the stability of the expansive soil slope can be reduced. However, few quantitative field studies have been carried out to investigate the influence of slope gradient on rainfall infiltration involving expansive soil slope. Studies on other types of soil slopes have yielded no strong consensus in the effect of inclination gradient on infiltration. Some researchers found there is no relationship between infiltration and slope gradient [15–17]. Others have reported a decrease in infiltration with increasing slope [18–20]. The relationship between the slope gradient and infiltration may change as the gradient increased [12,21,22]. The counterintuitive studies showed increased infiltration rate with increasing slope gradient [23–27].

In order to improve our understanding on the hydrological process for expansive soil slopes, providing a reference for the protection of expansive soil slopes, a well-instrumented field study was carried out on an unsaturated expansive soil slope nearby the proposed Guiyang-Nanning high-speed railway. The field test consisted of four neighboring monitoring sloping areas (2.1 m wide, 2 m high each) with different inclinations. Rainfall simulation tests and in situ monitoring were conducted. The data analyzed included the initiation time and amount of runoff, surface runoff rate, infiltration, changes of soil water content due to simulated rainfall. The influence of slope gradient and initial soil water content on runoff and infiltration was discussed by comparing the observed differences.

## 2. Materials and Method

### 2.1. Site Description and Characterization

The test site is located in the suburb of Nanning, Guangxi, China, near the Guiyang-Nanning high-speed rail line. The longitude and latitude are 108°23′ and 22°54′, respectively. The region has a subtropical monsoon climate, with a perennial average temperature of 21.8 °C and an average annual rainfall of about 1320 mm. A borehole investigation was carried out in the site before the test slopes were built. It was found that there is a significant depth of typical unsaturated expansive clay in the site. The geotechnical parameters were also obtained from the boreholes, as shown in Table 1. The groundwater depth was 12 m estimated from the borehole investigations. The influence of groundwater on the water content of surface soil could be neglected.

**Table 1.** Physical and mechanical properties of soil.

| Depth /m | Petrographic Description | Free Swell Ratio/% | Density /g cm⁻³ | Water Content/% | Dry Density/g cm⁻³ | Liquid Limit/% | Plastic Limit/% | Content of Clay and Fine Silt Less than 5 μm/% |
|---|---|---|---|---|---|---|---|---|
| 0–0.5 | Root soil | - | 1.76 | 14.70 | 1.53 | 34.2 | 18.3 | 22 |
| 0.5–1.2 | Reddish-brown expansive soil | 55 | 1.89 | 25.20 | 1.51 | 45.6 | 22.4 | 40 |
| 1.2–3.1 | Pale expansive soil | 98 | 1.97 | 28.90 | 1.53 | 77.8 | 29.1 | 64 |
| 3.1–4.0 | Mud rock | - | 2.08 | 18.50 | 1.76 | 34.3 | 16.2 | 24 |

The four neighboring experimental slopes were made into shape in July 2018, as shown in Figure 1. The slopes had the gradients of 70%, 47%, 32%, 21%, respectively, with the same height of 2 m and width of 2.1 m. The upper soil layer of test slopes was rare in cracks and fissures shortly after the molding. However, before the artificial rainfall test in May 2019, cracks of different degrees appeared in the surface due to the dry and wet circulation of the atmosphere. The steeper the slope, the more obvious were the cracks.

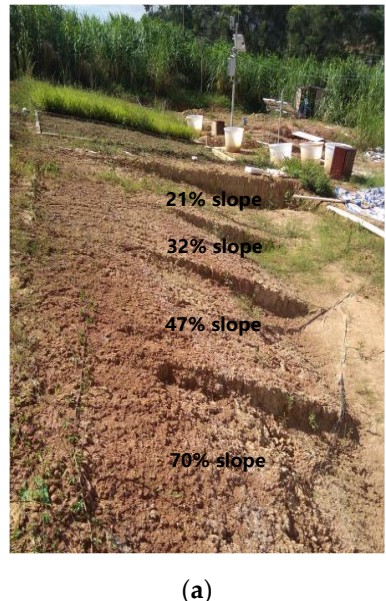

(**a**)

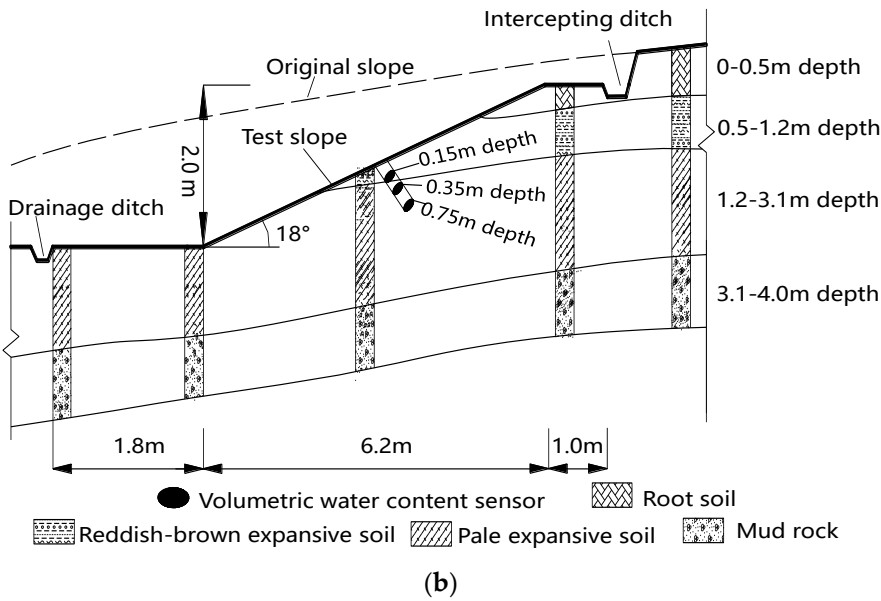

(**b**)

**Figure 1.** Experimental slopes: (**a**) Layout of four slopes; (**b**) cross-section of 32% slope.

The soil–water characteristic of the pale expansive soil sample at a depth of 1.5 m was obtained by the pressure plate instrument, and the experimental data was fitted using the VG soil–water characteristic curve model [28], as shown in Figure 2.

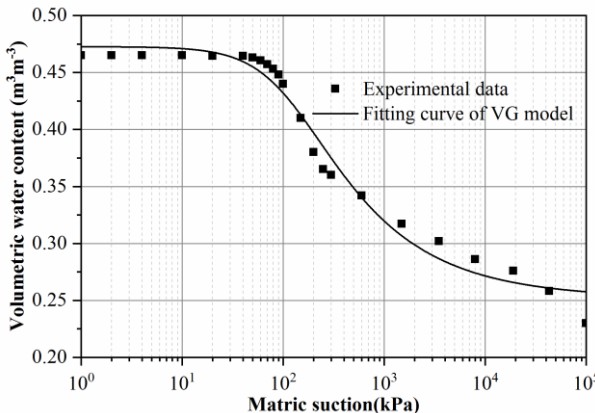

**Figure 2.** The soil–water characteristic curve of the expansive soil sample.

## 2.2. Test Instruments

The instruments included artificial rainfall simulation, runoff collection system, and moisture content sensors (EC-5).

Simulated rainfall was produced artificially using a sprinkler system comprised of a pump, a main water supply line, and nozzles (Figure 3a). Ten nozzles were installed on the retractable metal frame components, with a height of 3 m from the slope surface. The diameter of rainwater from each nozzle falling on the slope was about 1.0 m. The precipitation distribution was reasonable and even. The intensity of rainfall could be controlled effectively by the rotary knobs of spray units. A set of ten nozzles was found to be sufficient to produce the required rainfall on the slope.

A runoff collection trench was constructed on the slope and the surface runoff was measured using a scale barrel installed at the end of the trench, as shown in Figure 3b. The projected area of runoff collection plots in each slope is equal to the horizontal plane. Metal plates, 200 mm high and driven about 80 mm into the slope, were used to border the runoff to a lower plot. The boundaries guided the surface runoff into the scale barrel through a rectangular section of PVC pipe. The connection between the lower plot boundary and the PVC pipe was cemented to avoid any leakage. The barrel was cylindrical, 350 mm high and 200 mm in diameter, placed on the platform at the downstream end of the slope. A buffer plate was fixed inside the scale barrel, dampening turbulence, to give stable depth readings.

The moisture content sensors (EC-5) were buried in the middle of the slopes in January 2019. The layout of the sensors and data logger was shown in Figure 4. The moisture sensor needle was 60 mm in length and 5 mm in width. The measuring range of volumetric water content varied from 0 to 1 $m^3m^{-3}$ with the accuracy of ± 0.01. The measurement principle of the moisture sensor is based on the functional relationship between the dielectric constant and the moisture content of soil mass. Moisture sensors for each slope were mounted at the depths of 0.15, 0.35, and 0.75 m perpendicular to the slope, as shown in Figure 1b. After the installation of the sensors, the holes were backfilled with the soil dug out before. The backfill soil was compacted as far as possible to the in situ dry density, to prevent water from leaking into the slopes through the holes.

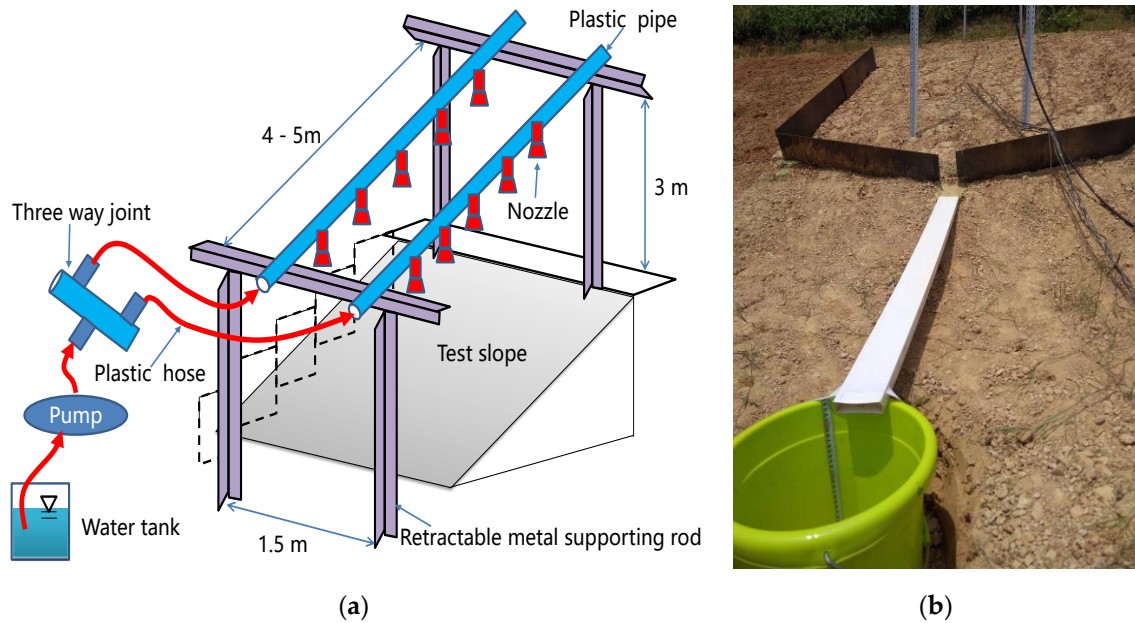

**Figure 3.** Equipment of simulated rainfall and runoff collection: (**a**) Sprinkler system; (**b**) runoff collection system.

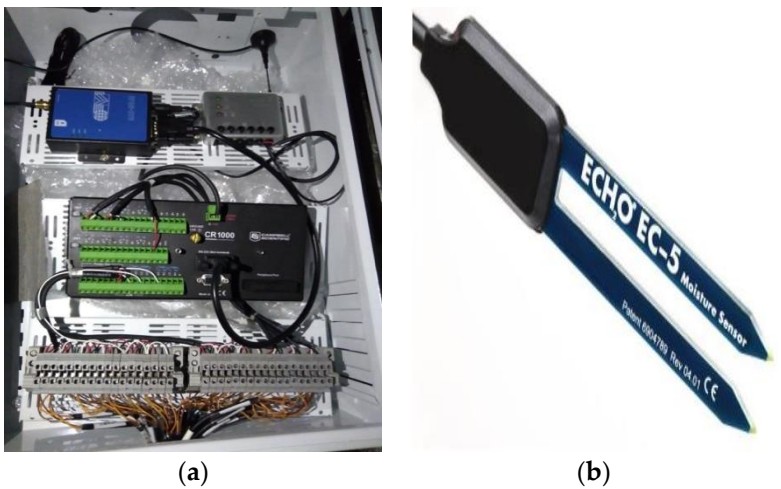

**Figure 4.** Data acquisition system and measuring sensor: (**a**) Data acquisition equipment; (**b**) volumetric water content sensor.

### 2.3. Experimental Design

The artificial rainfall test was carried out according to the initial soil moisture content of slope surface soil that was related to the previous natural rainfall condition. The simulated rainfall with the same intensity (48.4 mm/h) and duration (60 min) was made alternately at four slopes in the absence of natural rainfall. Before each test, five soil samples were taken within the range of 3 cm depth in the middle surface of each slope, and the mass moisture content of the samples was measured by the drying method. The average value of five soil samples was taken as the initial soil water content of the test.

The initial soil moisture and measured time of surface runoff in the artificial rainfall tests of each slope were shown in Table 2. Rainfall events with the initial soil water content of around 5%, 20%, and 31% were called "Dry run", "Moderate run", and "Wet run", respectively. First, the dry run was carried out, then the wet run, and after setting the moderate soil moisture the moderate run. The slopes and runoff collection system were covered with plastic cloth before each test. The pump was turned on after leveling the rainmaker. When the rainfall is stable, the plastic cloth on the

slope was removed immediately. At the same time, a stopwatch was used to record the time. During the rainfall, the initiation time of runoff at the outlet of the test slope was recorded. After the generation of runoff, the water depth scale value in the collection bucket was also recorded at regular intervals. The data acquisition instrument remained in working state to monitor the soil moisture with the interval of 1 min.

**Table 2.** Summary of runoff and infiltration of the slopes under simulated rainfall.

| | Area | | | Artificial Rainfall | | | | Runoff | | | | | Infiltration | | |
|---|---|---|---|---|---|---|---|---|---|---|---|---|---|---|---|
| Slope | Runoff Collection Area (m²) | Initial Moisture of Surface Soil (%) | Rainfall Event | Date | Duration (min) | Applied Precipitation During Runoff Monitoring (mm) | Monitor Time (min) | Start (min) | Measured Discharge Volume (mL) | Total/Area (mm) | Max. Rate (mm/h) | Final Coefficient (%) | Total/Area (mm) | Min. Rate (mm/h) | Final Coefficient (%) |
| 21% | 5.56 | 5.6 | Dry run | 24 May 2019 | 60 | 48.4 | 60 | 9.60 | 31,748 | 5.71 | 5.75 | 12.06 | 41.63 | 41.59 | 87.94 |
| 32% | 5.72 | 5.3 | | 22 May 2019 | | | | 8.08 | 50,679 | 8.86 | 14.50 | 19.25 | 37.17 | 31.53 | 80.75 |
| 47% | 6.00 | 5.7 | | 23 May 2019 | | | | 6.50 | 73,980 | 12.33 | 19.04 | 27.89 | 31.90 | 24.83 | 72.11 |
| 70% | 6.64 | 5.1 | | 24 May 2019 | | | | 2.75 | 59,162 | 8.91 | 13.16 | 22.39 | 30.90 | 26.49 | 77.61 |
| 21% | 5.56 | 19.8 | Moderate run | 25 June 2019 | 60 | 32.3 | 40 | 4.70 | 31,414 | 5.65 | 12.60 | 17.91 | 25.91 | 34.74 | 82.81 |
| 32% | 5.72 | 20.1 | | 25 June 2019 | | | | 3.95 | 41,527 | 7.26 | 15.34 | 23.67 | 23.42 | 30.70 | 76.33 |
| 47% | 6.00 | 20.2 | | 25 June 2019 | | | | 2.42 | 58,020 | 9.67 | 18.82 | 33.06 | 19.57 | 25.05 | 66.94 |
| 70% | 6.64 | 19.6 | | 25 June 2019 | | | | 1.63 | 53,585 | 8.07 | 14.94 | 30.52 | 17.38 | 24.71 | 69.48 |
| 21% | 5.56 | 32.1 | Wet run | 29 May 2019 | 60 | 16.1 | 20 | 1.57 | 44,202 | 7.95 | 27.80 | 50.41 | 7.83 | 19.54 | 49.59 |
| 32% | 5.72 | 31.8 | | 29 May 2019 | | | | 1.37 | 44,444 | 7.77 | 27.26 | 50.64 | 7.57 | 18.77 | 49.36 |
| 47% | 6.00 | 31.6 | | 29 May 2019 | | | | 1.13 | 49,140 | 8.19 | 27.88 | 55.46 | 6.58 | 15.98 | 44.54 |
| 70% | 6.64 | 31.3 | | 29 May 2019 | | | | 0.87 | 53,386 | 8.04 | 27.48 | 60.86 | 5.17 | 12.17 | 39.14 |

*2.4. Data Analysis*

The effect of soil erosion on the runoff that was not considered for the amount of soil erosion was small in this test. When rainfall duration is $t$ (min), the cumulative runoff per unit area for each test was calculated using the following formula:

$$R_t = \frac{10V}{S} \tag{1}$$

where:

$R_t$ —cumulative runoff per unit area (mm),
$V$ —runoff volume collected during $t$ time (mL),
$S$ —area of the runoff collection plot (cm²),
10—adjusting coefficient.

The instantaneous runoff rate can be calculated from:

$$r_i = \frac{600V_i}{St_i} \tag{2}$$

where:

$r_i$ —instantaneous runoff rate (mm h⁻¹),
$V_i$ — $i\,th$ runoff volume collected (mL),
$t_i$ —interval time to collect runoff samples (min).

Suppose the rainfall duration is $t$ (min). The cumulative infiltration per area for each test was calculated by the following formula:

$$F_t = \frac{P\cos\alpha t}{60} - R_t \tag{3}$$

where:

$F_t$ —cumulative infiltration per area (mm),
$P$ —rainfall intensity (mm h⁻¹),
$\alpha$ —slope (°).

The instantaneous infiltration rate was calculated by the following formula:

$$f_i = P\cos\alpha - r_i \tag{4}$$

where:

$f_i$ —instantaneous infiltration rate (mm h⁻¹).

The infiltration coefficient represents the rainwater that seeps into the slope as a percentage of the total rainfall, which can be calculated by the following formula:

$$c_F = 1 - \frac{60R_t}{P\cos\alpha t} \tag{5}$$

where:

$c_F$ —infiltration coefficient (dimensionless).

## 3. Results and Discussion

### 3.1. Surface Runoff

#### 3.1.1. Runoff Initiation Time

In May and July 2019, several simulated rainfall tests were carried out in the site to study the influence of slope gradient and initial moisture content on runoff and infiltration for expansive soil slopes. The runoff and infiltration of the slopes under simulated rainfall were summarized in Table 2.

In the early stage of rainfall, all the rainwater falling on the slope seeped into the soil. As the rainfall continued, the infiltration potential gradient of surface soil decreased after absorbing water, and surface cracks sealed for soil expansion, which resulted in the decrease in the infiltration capacity of soil. If the infiltration capacity of soil is less than the rainfall rate, the excess rainwater will flow down the slope. Some of them will infiltrate into the soil downstream of the slope, and the other part will completely flow out of the slope, forming runoff. Runoff initiation time is defined as the time from the beginning of rainfall to the observation of runoff generation in the runoff collection bucket. According to the initial water content of the surface soil during the test, the runoff initiation time for each slope under simulated rainfall events was shown in the following histogram (Figure 5).

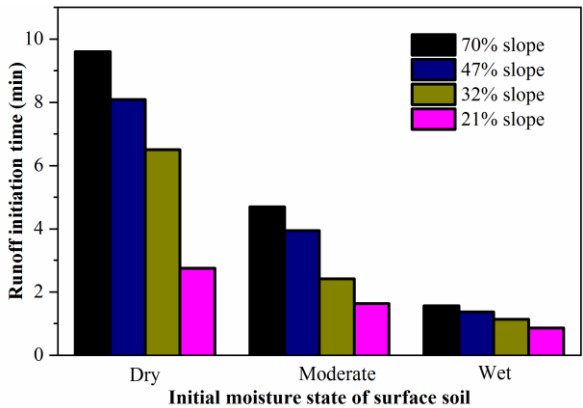

**Figure 5.** Runoff initiation time of each slope under different initial soil water contents.

It can be seen from Figure 5 that runoff initiation time was related to the slope gradient and the initial water content of surface soil. Under the same initial moisture condition of surface soil, the runoff initiation time decreased with increasing slope gradient. For the same slope, the higher the initial water content, the earlier the runoff started.

#### 3.1.2. Cumulative Runoff per Unit Area

The cumulative runoff per unit area of each slope under different rainfall events was shown in Figure 6. As can be seen from Figure 6, the cumulative runoff per unit area of each slope showed a roughly linear growth trend with the duration of rainfall. The greater the initial water content of surface soil, the less obvious the influence of slope on the cumulative runoff per unit area. In the Dry run (Figure 6a), the cumulative runoff per unit area grew slowly with time in the initial stage after runoff generation, and the larger the slope gradient was, the grater the cumulative runoff per unit area was in the same rainfall duration. With the continuation of rainfall (after 25 min), the cumulative runoff per unit area of 47% slope exceeded 70% slope, while that of 32% slope was lower than 70% slope, but the difference gradually diminished. At the end of rainfall, the cumulative runoff per unit area of 32% slope tended to exceed 70% slope. The cumulative runoff per unit area of 21% slope was always at the lowest level. In the Moderate run (Figure 6b), the greater the slope was, the greater the cumulative runoff per unit area would be if the slope is less than 47%. When the slope exceeded 47%, the cumulative runoff did not increase with increasing slope. In the Wet run (Figure

6c), the cumulative runoff per unit area of each slope had little difference, and the influence of slope on the cumulative runoff per unit area can be ignored.

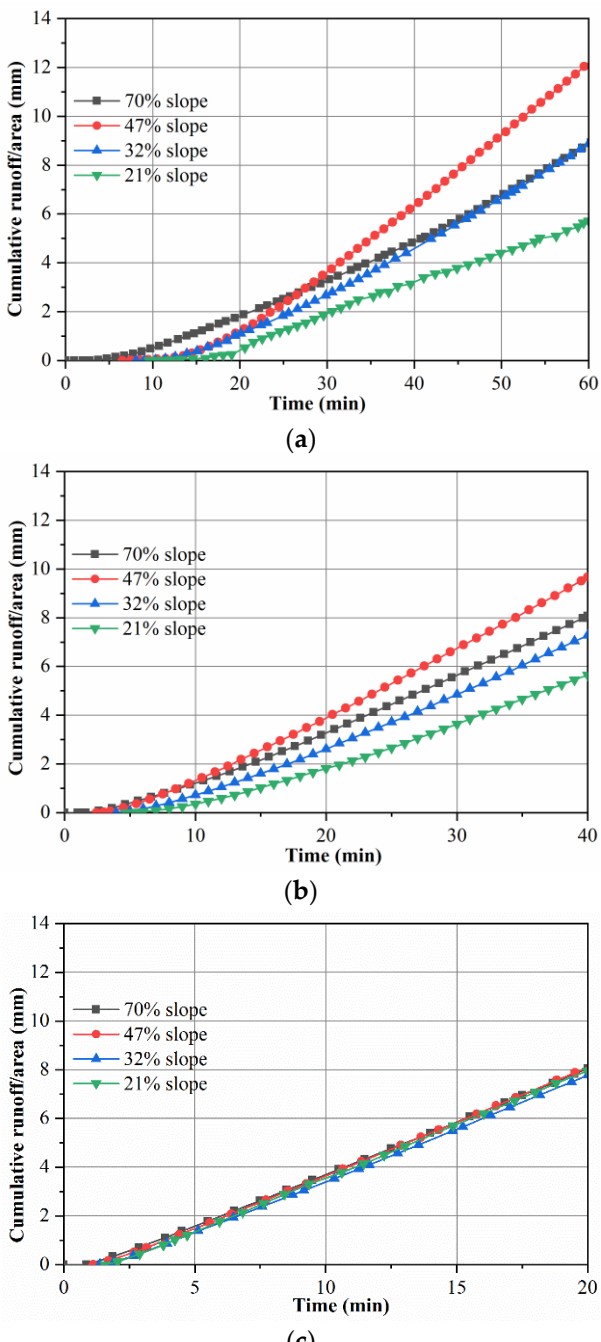

**Figure 6.** The cumulative runoff per unit area as a function of time of each slope under different rainfall events: (**a**) Dry run; (**b**) Moderate run; (**c**) Wet run.

It can be seen that there is no definite rule about the influence of slope gradient on the cumulative runoff per unit area in expansive soil slopes. The cumulative runoff per unit area is mainly determined by the permeability characteristics of soil and the rainfall per unit area of slope in rainfall with the same intensity. Soil permeability and rainfall per unit area are related to slope gradient. The larger the slope gradient is, the more seriously the slope is affected by the dry and wet cycle, and the more developed the cracks of the surface soil [14], so that the permeability of soil tends to increase with increasing slope. However, the larger the slope gradient, the smaller is the gradient of infiltration potential generated by gravity [29], so the relationship between permeability and slope tends to be complex. On the other hand, slope gradient affects the rain-bearing area of slope. When

the projected area of the slopes on the horizontal plane is equal, the larger the slope is, the larger the rain-bearing area and the smaller the rainfall per unit area, leading to the trend that the runoff per unit area decreases with increasing slope gradient. When the initial water content of the surface soil is close to saturation, the cracks in the surface soil of each slope are sealed, and the difference of soil permeability caused by the cracks is minimal. However, the increase or decrease of runoff caused by the gradient of infiltration potential energy and rainfall per unit area of slope may offset each other, so the runoff difference of each slope is small. However, the relationship between cumulative runoff per unit area and slope gradient is more complex if the initial moisture content of soil is in a relatively dry state. When the slope is more than 47%, the difference of soil permeability characteristics caused by fissures plays a dominant role in the influence of runoff, and the cumulative runoff per unit area tends to decrease with the increasing slope.

### 3.1.3. Runoff Rate

The relationship between the runoff rate of each slope and the duration of rainfall was shown in Figure 7. As can be seen from Figure 7, almost all rainwater penetrated into the soil and no runoff was observed in the early stage of rainfall. Runoff of each slope occurred after a certain period of rainfall. Fitting analysis (Table 3) showed that the runoff rate in the Dry run and Moderate run presented a logarithmic upward trend with the duration of rainfall, and the determination coefficients were all greater than 0.90. With the increase of time, the growth rate gradually decreased. For the same slope, the lower the initial soil moisture content, the slower the increase rate of runoff rate in the initial stage, and the longer the time needed to reach the stable state of runoff.

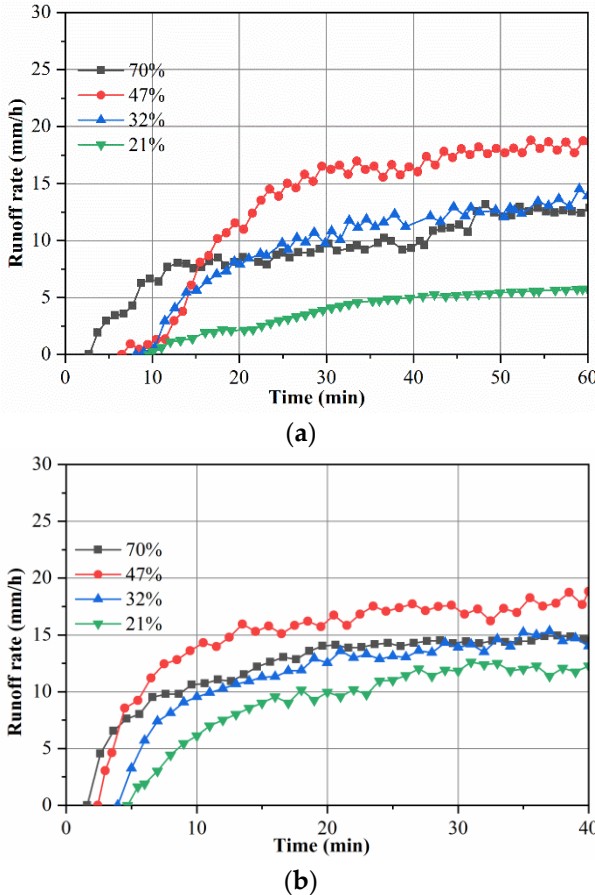

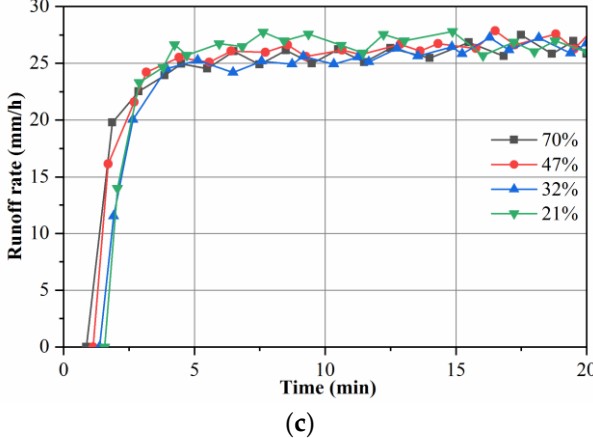

(**c**)

**Figure 7.** Runoff rate as a function of time on each slope under different rainfall events: (**a**) Dry run; (**b**) Moderate run; (**c**) Wet run.

**Table 3.** Fitted equations of runoff rate with time.

| Slope | Rainfall Event | Fitted Equation | $R^2$ |
|---|---|---|---|
| 21% | | $r_i = 3.2378 \ln t - 7.2012$ | 0.9808 |
| 32% | | $r_i = 6.4961 \ln t - 12.200$ | 0.9474 |
| 47% | Dry run | $r_i = 9.3171 \ln t - 17.768$ | 0.9190 |
| 70% | | $r_i = 3.7144 \ln t - 2.8647$ | 0.9342 |
| 21% | | $r_i = 5.4520 \ln t - 6.7938$ | 0.9474 |
| 32% | | $r_i = 3.9643 \ln t + 1.1009$ | 0.9443 |
| 47% | Moderate run | $r_i = 5.2446 \ln t + 0.0197$ | 0.9054 |
| 70% | | $r_i = 5.1732 \ln t - 3.3974$ | 0.9174 |

In the Dry run (Figure 7a), the larger the slope gradient was, the higher the runoff rate in the initial stage. With the continuous rainfall, the runoff rate of 47% slope exceeded 70% slope, becoming the largest when the rainfall lasts for 15 min. The runoff rate of 32% slope was lower than that of 70% slope at the initial stage, and was close to that of 70% slope after 21 min. The runoff rate of 21% slope had always been the lowest. The change of runoff rate for the Moderate run (Figure 7b) was similar to that for the Dry run. In the Wet run (Figure 7c), once runoff occurred on each slope, the runoff rate increased rapidly and soon reached a stable state (about 3 min). There was little difference in the runoff growth trend and the stable runoff rate of each slope, and the slope gradient had little impact on the runoff rate. The Wet run often resulted in a steeper runoff rising limb and earlier and higher runoff peak compared with the Dry run, because soil cracked and subsequently a large amount of infiltration occurred in the Dry run. One could perhaps go more closely into the cracks and swelling and their dynamics.

*3.2. Infiltration*

3.2.1. Cumulative Infiltration per Unit Area

The relationship between the cumulative infiltration per unit area of each slope and the duration of rainfall was shown in Figure 8. The cumulative infiltration per unit area was calculated by subtracting the cumulative runoff per unit area from the cumulative rainfall per unit area when the rainfall loss caused by interception and evaporation is ignored. It can be seen from Figure 8 that, the larger the slope gradient was, the smaller the cumulative infiltration per unit area, regardless of the initial soil water content. The cumulative infiltration per unit area of each slope had a significant linear growth relationship with rainfall duration. For the same slope, the higher the initial soil moisture content was, the slower the increase rate of cumulative infiltration per unit area with the duration of rainfall. The cumulative infiltration per unit area of expansive soil slope decreased

significantly with increasing slope and initial soil moisture content, which was consistent with the conclusion of Huang et al. [30]. The increase was much higher with the premoisture than with the slope inclination.

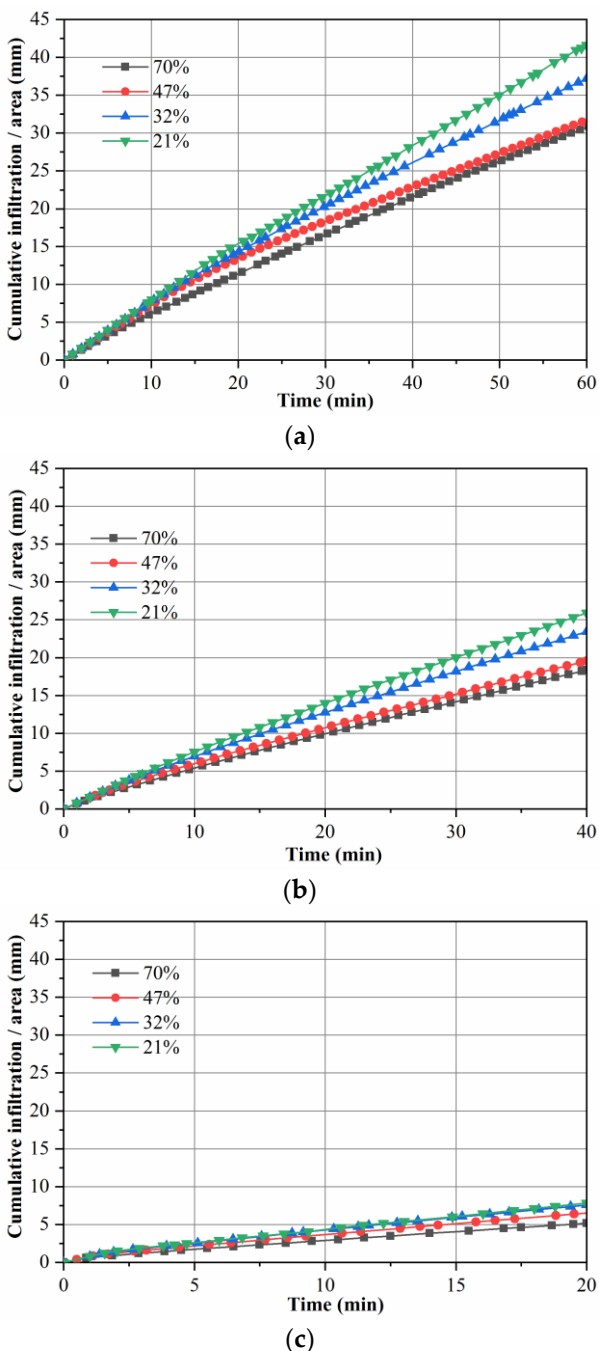

**Figure 8.** Cumulative infiltration per unit area as a function of time of each slope under different rainfall events: (**a**) Dry run; (**b**) Moderate run; (**c**) Wet run.

3.2.2. Infiltration Coefficient

Infiltration coefficient is an important parameter to evaluate the efficiency of rainfall infiltration. It represents the percentage of rainfall converted into seepage in total rainfall and indirectly describes the percentage of rainfall lost by runoff. The relationship between infiltration coefficient and rainfall duration of each slope under different rainfall events was shown in Figure 9.

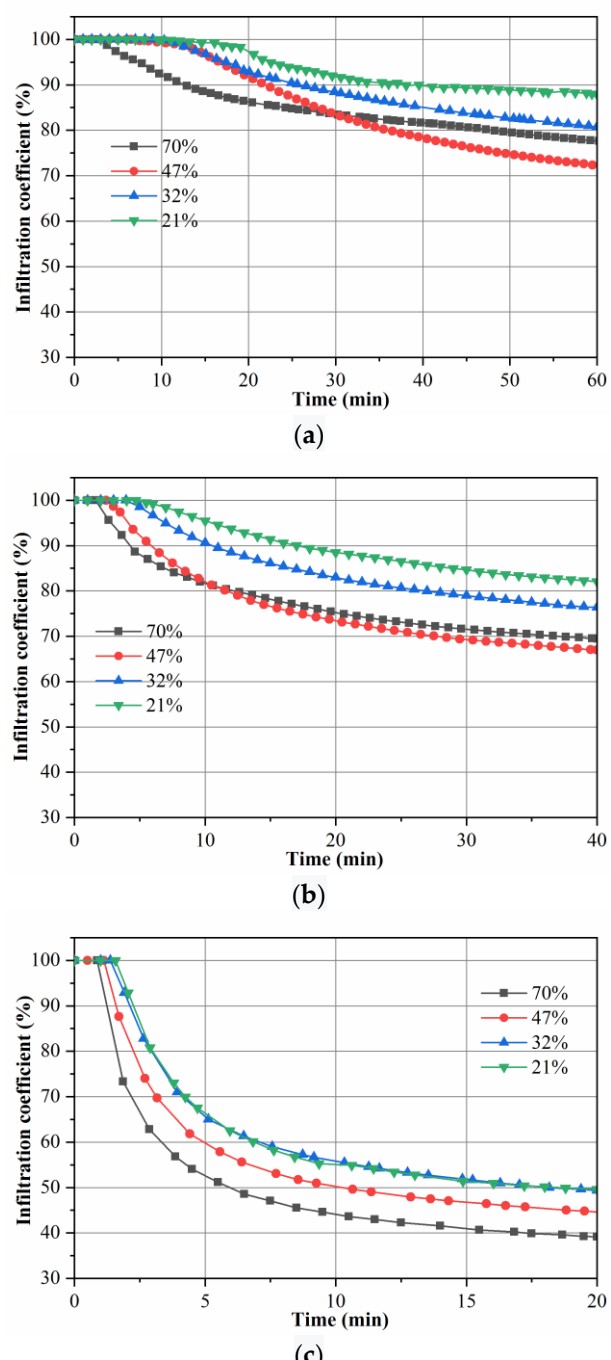

**Figure 9.** Infiltration coefficient as a function of time of each slope under different rainfall events: (**a**) Dry run; (**b**) Moderate run; (**c**) Wet run.

It can be seen from Figure 9 that in the early stage of rainfall, all rainwater penetrated before the occurrence of runoff, and the infiltration coefficient was equal to 100%. After the occurrence of runoff, the infiltration coefficient kept decreasing. With the continuous rainfall, the rate of infiltration coefficient gradually tended to be stable. The larger the initial soil moisture content of the same slope, the lower the infiltration coefficient at the same time, and the shorter the time to reach the stable infiltration coefficient. Infiltration accounted for a large proportion of rainfall, with more than 65% in both Dry run and Moderate run. The least stable infiltration coefficient existed in the wet run of 70% slope, but it was also over 39%.

The model test of Huang et al. [30] showed that the infiltration coefficient decreased with the increasing rainfall intensity, slope and initial soil moisture content, and the slope was considered as the least important influencing factor. The field test of Chen et al. [31] concluded that slope gradient

played an important role in infiltration recharge. The difference may be caused by the sealing degrees of slope surface in the model test and field test. Ran et al. [32] concluded that the surface seal should not be ignored when estimating the runoff and seepage. If the expansive soil slope is exposed to the atmosphere, the dry and wet cycle led to the crack of slope surface. Due to the limitations of instruments and equipment, the detailed measurement of the crack development process of each slope was not possible. Therefore, this study cannot quantitatively analyze the impact of crack development pattern on rainfall infiltration. However, it can still be found by slope surface morphology on site before the Moderate run (Figure 10) that the larger the slope gradient was, the more seriously the expansive soil slope was affected by the dry and wet cycle, and the lower the initial sealing degree of slope surface was. The decrease in the infiltration coefficient was primarily related to the open cracks and fissures that had become filled and gradually sealed because of the swelling of the soil upon wetting. Another reason may be due to the decrease of the hydraulic gradient as a result of reduction in soil suction.

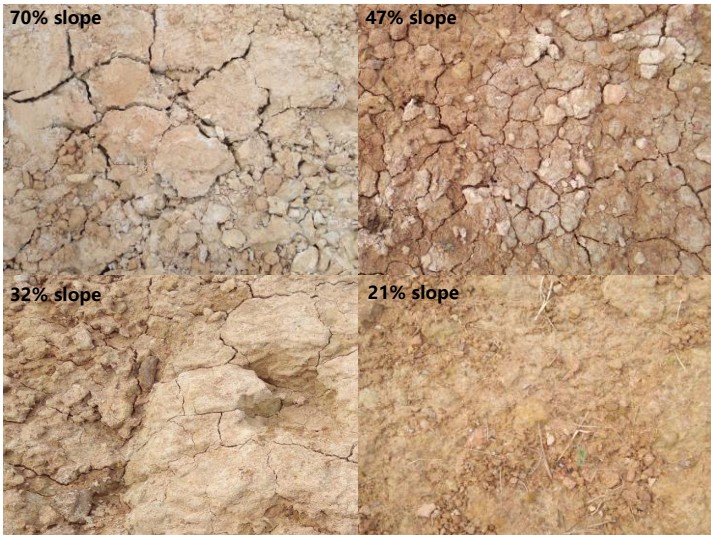

**Figure 10.** Photos of slope surface morphology before Moderate run.

In this test, the influence of slope gradient on infiltration coefficient was obvious. In the Dry run and Moderate run, the infiltration coefficient at the same time decreased with increasing slope gradient when the slope is less than 47%. However, the infiltration coefficient of 70% slope was greater than that of 47% slope. There may be a critical slope between 47% and 70%. If the slope is smaller than the critical slope, the infiltration coefficient decreases with increasing slope. However, the infiltration coefficient increases with increasing slope if the slope is larger than the critical slope. The infiltration coefficient at the same time decreased with increasing slope in the Wet run. The infiltration coefficient of small slope is greater than that of larger slope when the permeability of the soil is equal. However, it is affected by the high permeability caused by cracks when soil is dry.

### 3.3. Water Content Changes Due to Simulated Rainfall

The change in volumetric water content measured by EC-5 sensors at three different depths of each slope in simulated rainfall for the Dry run was shown in Figure 11. The sensor at 35 cm depth in 47% slope failed to collect data. The moisture content of soil at 15 and 35 cm depth in 70% slope and 15 cm depth in 47% slope changed with the duration of rainfall significantly.

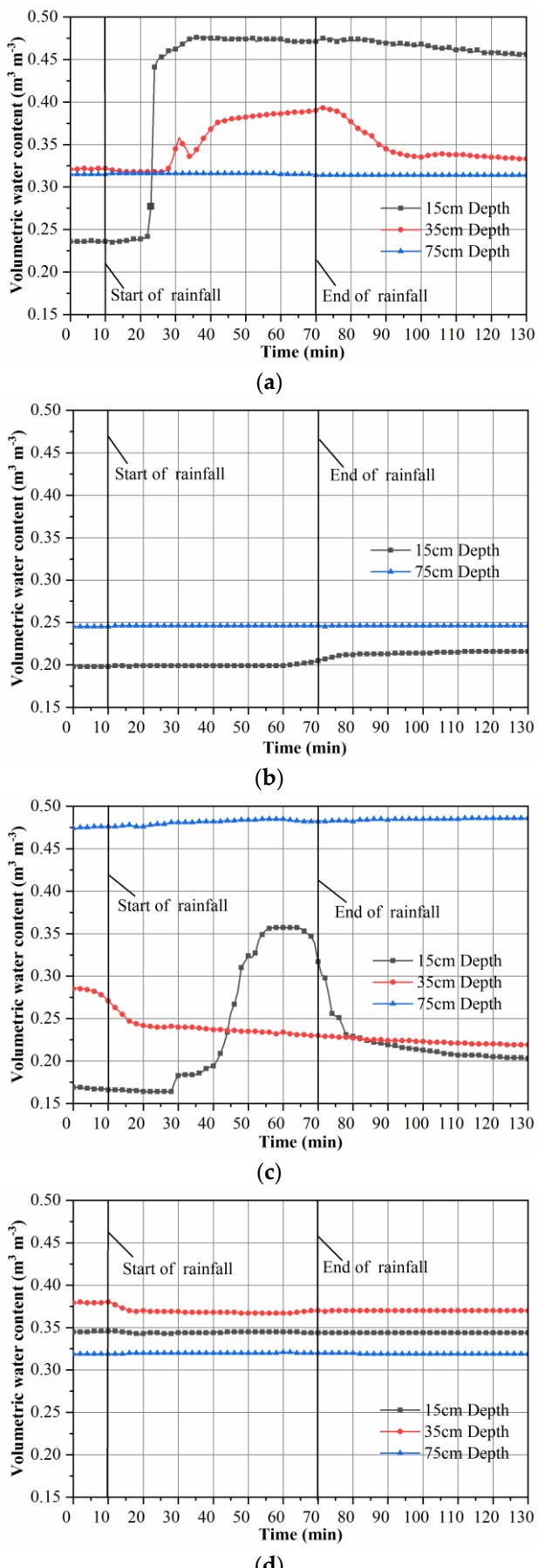

**Figure 11.** Response of volumetric water contents in the Dry run: (**a**) 70% slope; (**b**) 47% slope; (**c**) 32% slope; (**d**) 21% slope.

The moisture content sensor at 15 cm depth in 70% slope started to respond when it rains for about 10 min. Then, the volumetric water content of soil at this depth increased rapidly and reached a stable value close to the saturation, remaining at the same level 60 min after the rainfall stopped. According to the monitoring data in the later period, the water content of the soil in this depth did not decrease until 6 h after the rainfall stopped. It indicates that the preferential flow caused by cracks exists at about 15 cm depth in 70% slope, which makes the soil response to rainfall more quickly and shows obvious perched water characteristics after the rainfall stops. It is consistent with the conclusion of Stewart R. D. et al. [14] that crack networks play a dominant role in controlling the water movement in a vertical soil and shrinkage cracks cause surface water to reach depths faster than the infiltration through the soil matrix. The moisture content of soil at 35 cm depth in 70% slope began to increase gradually at about 18 min after rainfall, and the increasing rate was slow. It kept increasing and did not reach the stable value of saturated water content at the end of rainfall. It decreased gradually after the rainfall stops for 5 min and no perched water was formed. The moisture content of soil at 15 cm depth in 47% slope showed a slight increase after 55 min of rainfall. The rainfall response of soil at 15 cm depth in 32% slope was consistent with that of the soil at 35 cm depth in 70% slope. The moisture content of soil at three depths in 21% slope did not increase during the simulated rainfall test.

From the experimental results, the greater the slope, the faster the rainwater penetrates into the soil, and the larger the infiltration capacity of surface soil is. Theoretically speaking, the larger the slope is, the smaller the hydraulic gradient generated by gravity, and the smaller the permeability of soil with the same physical properties. In fact, the larger the slope is, the greater the influence of dry and wet cycle on the surface of expansive soil slopes, the faster the development of cracks in the slope, and the smaller the sealing degree of the slope, which has a far greater impact on the soil permeability than gravity.

The response of slope to rainfall may change because partial cracks may seal after soil expanding upon absorbing water. In the Moderate run, the moisture content of soil at the three depths of each slope did not change except for the apparent increase of the moisture content measured at the depths of 15 and 35 cm in 70% slope and the depth of 15 cm in 47% slope. In the Wet run, the moisture content sensors at three depths of each slope did not respond. The change of soil water content caused by rainfall is influenced by rainfall characteristics and climate before the rainfall event [33]. The initial soil moisture content is large if there is much rainfall before the artificial rainfall test. The permeability of expansive soil is low due to the high sealing degree of slope surface. Therefore, it was difficult for the rainwater to penetrate into the depth of 15 cm below the slope surface when the rainfall lasts for 60 min with the intensity of 48.4 mm/h. Moisturizing measures can be taken for expansive soil slopes to reduce cracks caused by the dry and wet cycle, which prevents the rainwater from infiltrating into the slope.

## 4. Conclusions

Based on a series of simulated rainfall experiments on expansive soil slopes with different inclinations, the following conclusions can be drawn.

The initiation time of runoff, runoff rate, infiltration coefficient, and the increment of soil water content caused by rainfall on expansive soil slopes were affected by the inclination gradient and especially by the initial soil moisture content. In the early stage, all the rainwater falling on the expansive soil slope infiltrated into the soil mass. With the continuous rainfall, the infiltration capacity of the slope soil mass decreased, and some of the rainwater formed runoff on the slope surface. The initiation time of runoff decreased with the increasing slope and initial soil water content.

After the generation of runoff, the cumulative runoff per unit area increased linearly with time. The effect of slope on cumulative runoff per unit area decreased with the increasing initial soil moisture content. The slope had little effect on cumulative runoff per unit area when the initial soil water content approaches saturation.

The runoff rate increased logarithmically with time. In the early stage, the runoff rate increased the fastest. With the continuous rainfall, the runoff rate grew slowly, and then tended to be stable in fluctuation. Compared with the Dry run, the Wet run often resulted in a steeper runoff rising limb, earlier and higher runoff peak. The runoff rate at the same rainfall duration did not always increase with the increasing slope. There may be a critical slope gradient between 47% and 70%.

The cumulative infiltration per unit area of each slope increased linearly with the duration of rainfall and decreased with the increasing initial soil moisture content and slope. A rainfall with an intensity of approx. 50 mm/h may contribute from about 39% to 100% of its total rainfall as infiltration. After runoff occurs, the infiltration coefficient decreased rapidly and tended to be stable gradually. The smaller is the initial soil moisture content for the same slope, the longer the time to reach the stable infiltration coefficient, the higher the percentage of rainfall contributing to infiltration. Slope gradient affected the crack morphology, infiltration potential energy gradient, and rainfall per unit area in expansive soil slopes, and thus had a significant impact on the infiltration coefficient. With the increase of the initial soil moisture content, the effect of slope on infiltration coefficient decreased.

The smaller the slope and the higher the initial soil moisture content, the weaker the change of soil water content caused by simulated rainfall. Regardless of the initial soil moisture content, the rainwater was hard to infiltrate to 15 cm depth of the 21% slope. It was difficult for the simulated heavy rain to penetrate into the depth of 15 cm below the slope surface quickly in the Moderate and Wet run.

The characteristics of runoff generation, infiltration processes, and volumetric water content changes identified in this study may be useful for understanding the seepage characteristics of expansive soil slopes due to rainfall and have relevance to the protection method of the slope based on surface water content control. It would be interesting to complement such studies in terms of preferential flow paths and their dynamics within expansive soils.

**Author Contributions:** Conceptualization, G.M.; Methodology, W.L. and H.L.; Validation, P.C. and H.D.; Formal analysis, G.M.; Investigation, P.C. and H.D.; Resources, G.M.; Data curation, P.C.; Writing—original draft preparation, W.L.; Writing—review and editing, W.L. and L.F.; Visualization, L.F.; Supervision, H.L.; Project administration, W.L.; Funding acquisition, G.M. All authors have read and agreed to the published version of the manuscript.

**Funding:** This work was supported by the National Natural Science Foundation of China (Grant No. 41672296) and the Innovation Project of Guangxi Graduate Education (Grant No. YCSW2019046).

**Conflicts of Interest:** The authors declare no conflict of interest.

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
