# Peer review of "Study on Runoff and Infiltration for Expansive Soil Slopes in Simulated Rainfall"

_water, doi:10.3390/w12010222_

Round 1

Reviewer 1 Report

9 The management of such soils is hardly dealt with in the article, so it is better to omit it in the abstract.

14 It is recommended to mention the more important influence first. “The runoff initiation time decreased with initial soil water content and increasing slope gradient.”

18 contribute instead of contributed

20 Like 14 mention the more important influence first.

In the abstract, it is also recommended to refer to the relationship to observed cracks and the contradictory results of infiltration and soil moisture reaction. It should also be noted in the abstract that the influence of inclination is no longer discernible at high initial moisture levels.

Please pay attention to uniform citation methods

35 Charles W.W. Ng et al.

36 Zhan et al.

40 Qi & Vanapalli et al.

42 Khan, M. S. et al.

57 Robert E. Horton

90 Clay is defined as grain size < 2µm: Content of clay and fine silt less than 5μm

92 Change punctuation 70%, 47%, 32%, 21%

188 It would be helpful to integrate into table 2 the applied precipitation and the measured discharge volume.

248 … after a certain period of rainfall.

270 One could perhaps go more closely into the cracks and swelling and their dynamics.

283 The increase was much higher with the pre-moisture than with the slope inclination.

317 It might be helpful to explain earlier in the text that first the dry run was carried out, then the wet run and after setting moderate soil moisture the moderate run.

332 From Figure 10 one would conclude that (with the same moisture) the larger the slope inclination, the larger the cracks. From this one could conclude: larger cracks, more infiltration. The tests show the opposite. This contradiction should be clearly emphasised

379 Like before 332. Results from infiltration measurements, photos of slope surface morphology and soil moisture measurements seem to contradict each other. The infiltration measurement shows the highest infiltration with dry preconditions and low slope inclination. Because the photos of the slope surface morphology show the highest sealing and lowest crack formation at low slope inclination and because the soil moisture sensors show no reaction, the opposite would be assumed. This contradiction should be clearly emphasised.

386 “…were affected by the gradient and especially by the initial soil moisture content.”
It could be worked out more explicitly how much influence the initial moisture has (5%;20%;30%) and how much influence the slope inclination has?

399 Critical of what? Because with increasing slope inclination more rain infiltrates the soil again and the steep slopes become unstable?

402 “A rainfall with an intensity of approx. 50mm/h may contribute…”

411 “Regardless of the initial soil moisture content, the rainwater was hard to infiltrate into the 21% slope.” This contradicts the best cumulative infiltration (a) 88%; (b) 83%; (c) 50%. Where does the water flow? It may flow past soil moisture sensors in preferred flow paths.

416 It would be interesting to complement such studies in terms of preferential flow paths and their dynamics within expansive soils.

Author Response

Dear reviewer, Thanks for your review of my article. Your comments and suggestions have been very helpful in improving my paper. I have made the following modifications to the paper according to your suggestions. May we kindly ask you to review it again and give us your valuable comments? Best regards, Wenkai Lei Responses to reviewer 9 The management of such soils is hardly dealt with in the article, so it is better to omit it in the abstract. The management of such soils has been omitted from the abstract. 14 It is recommended to mention the more important influence first. “The runoff initiation time decreased with initial soil water content and increasing slope gradient.” It has been changed as recommended. 18 contribute instead of contributed It has been corrected. 20 Like 14 mention the more important influence first. It has been changed as recommended. In the abstract, it is also recommended to refer to the relationship to observed cracks and the contradictory results of infiltration and soil moisture reaction. It should also be noted in the abstract that the influence of inclination is no longer discernible at high initial moisture levels. Corrections have been made in the abstract. The larger the initial sealing degree of slope surface was,the smaller the cumulative infiltration per unit area of the slope. But the soil moisture reaction was more obvious. The influence of inclination is no longer discernible at high initial moisture levels. Please pay attention to uniform citation methods 35 Charles W.W. Ng et al. 36 Zhan et al. 40 Qi & Vanapalli et al. 42 Khan, M. S. et al. 57 Robert E. Horton The above contents and the format of References have been corrected. 90 Clay is defined as grain size < 2µm: Content of clay and fine silt less than 5μm Correction has been made in Table 1. 92 Change punctuation 70%, 47%, 32%, 21% The punctuations above have been corrected 188 It would be helpful to integrate into table 2 the applied precipitation and the measured discharge volume. The applied precipitation and the measured discharge volume have been integrated into table 2. 248 … after a certain period of rainfall. It has been corrected. 270 One could perhaps go more closely into the cracks and swelling and their dynamics. It has been added. 283 The increase was much higher with the pre-moisture than with the slope inclination. It has been added. 317 It might be helpful to explain earlier in the text that first the dry run was carried out, then the wet run and after setting moderate soil moisture the moderate run. The revised explanation was made in 2.3. Experimental design. 332 From Figure 10 one would conclude that (with the same moisture) the larger the slope inclination, the larger the cracks. From this one could conclude: larger cracks, more infiltration. The tests show the opposite. This contradiction should be clearly emphasized. Under the same moisture condition, the infiltration amount per unit area of the slope is determined by the precipitation per unit area, hydraulic gradient and soil permeability. The slope determines the precipitation per unit area and hydraulic gradient. The larger the slope is, the smaller the precipitation per unit area and hydraulic gradient. The cracks determine the permeability coefficient of soil. Larger cracks, larger the initial permeability coefficient of soil. But the open cracks and fissures may become filled and sealed because of the moisture of the expansive soil upon wetting. So the initial larger cracks can not indicate the final infiltration must be greater. 379 Like before 332. Results from infiltration measurements, photos of slope surface morphology and soil moisture measurements seem to contradict each other. The infiltration measurement shows the highest infiltration with dry preconditions and low slope inclination. Because the photos of the slope surface morphology show the highest sealing and lowest crack formation at low slope inclination and because the soil moisture sensors show no reaction, the opposite would be assumed. This contradiction should be clearly emphasised. The cumulative infiltration amount per unit area of a steeper slope is smaller than lower slope. But the moisture sensors in steeper slope show obvious reaction and the moisture sensors in lower slope show no reaction. It seems contradictory, but it is not. The steeper has larger cracks, and water may infiltrate into greater depths in preferred flow paths. The gentle slope has lower crack, and water infiltration in slope soil is uniform and slow. Under the condition that the rainfall with an intensity of approx. 50mm/h lasts for 1h, the water cannot infiltrate to a depth of 15cm. 386 “…were affected by the gradient and especially by the initial soil moisture content.” It has been added. It could be worked out more explicitly how much influence the initial moisture has (5%;20%;30%) and how much influence the slope inclination has? It makes sense to work out more explicitly how much influence the initial moisture and the slope inclination has. This may involve statistical analysis, which is a difficult job for me. 399 Critical of what? Because with increasing slope inclination more rain infiltrates the soil again and the steep slopes become unstable? The cumulative infiltration per unit area of the slope can be reduced by increasing the slope slope appropriately. However, the slope should not be too large. After exceeding the critical slope, the cumulative runoff per unit area of the slope increases instead. 402 “A rainfall with an intensity of approx. 50mm/h may contribute…” It has been corrected. 411 “Regardless of the initial soil moisture content, the rainwater was hard to infiltrate into the 21% slope.” This contradicts the best cumulative infiltration (a) 88%; (b) 83%; (c) 50%. Where does the water flow? It may flow past soil moisture sensors in preferred flow paths. The statement is incorrect and has been corrected as follows. Regardless of the initial soil moisture content, the rainwater was hard to infiltrate to 15cm depth of the 21% slope. The gentle slope has lower crack, and water infiltration in slope soil is uniform and slow. Under the condition that the rainfall with an intensity of approx. 50mm/h lasts for 1h, the water cannot infiltrate to a depth of 15cm. 416 It would be interesting to complement such studies in terms of preferential flow paths and their dynamics within expansive soils. It has been added.

Reviewer 2 Report

Dear authors,

Thank you for your submission. I find this article to be quite excellent. Your study on expansive soil slopes is consistent and it makes an advance in soil water dynamics studies. I just suggest some small recommendations:

-Some management measures are proposed in the paragraph from line 45 to line 53. Some citations are missing here.

-Unclear syntax in line 160. Please rewrite this sentence.

-In Conclusions, I miss some specific management recommendations related to erosion risk in expansive soil slopes.

Author Response

Dear reviewer, Thanks for your review of my article. Your comments and suggestions have been very helpful in improving my paper. I have made the following modifications to the paper according to your suggestions. May we kindly ask you to review it again and give us your valuable comments? Best regards, Wenkai Lei Responses to reviewer -Some management measures are proposed in the paragraph from line 45 to line 53. Some citations are missing here. References 9-12 have been added -Unclear syntax in line 160. Please rewrite this sentence. This sentence has been rewritten as following. Suppose the rainfall duration is (min). The cumulative infiltration per area for each test was calculated by formula -In Conclusions, I miss some specific management recommendations related to erosion risk in expansive soil slopes. The risk of erosion can be reduced by pre-moisture the expansive soil slope.

Round 2

Reviewer 1 Report

The revision has been carried out satisfactorily and has led to better legibility and understanding.